# Numerical Study on Elastic Buckling Behavior of Diagonally Stiffened Steel Plate Walls under Combined Shear and Non-Uniform Compression

**Yuqing Yang \*, Zaigen Mu and Boli Zhu**

School of Civil and Resource Engineering, University of Science and Technology Beijing, Beijing 100083, China; zgmu@ces.ustb.edu.cn (Z.M.); zboly@ustb.edu.cn (B.Z.)
\* Correspondence: yqyang@ustb.edu.cn

**Abstract:** Unstiffened steel plate walls (SPWs) are prone to buckling in practical engineering and will invariably be subjected to vertical loads. The use of stiffeners can improve the buckling behavior of thin plates. Considering the effect of the torsional stiffness of C-shaped stiffeners, the elastic buckling of the diagonally stiffened steel plate wall (DS-SPW) under combined shear and non-uniform compression is investigated. The interaction curves for the DS-SPW under combined action are presented, as well as a proposed equation for the elastic buckling coefficient. In addition, the effects of the stiffener's flexural and torsional stiffness on the elastic buckling stress were investigated, and the threshold stiffness formulae were proposed. The results show that the interaction curve of the DS-SPW under combined shear and non-uniform compression is approximately parabolic. The critical buckling stress of the DS-SPW can be increased by increasing the stiffener's torsional-to-flexure stiffness ratio and the non-uniform compression distribution factor, while the buckling stress can be decreased by increasing the non-uniform compression-to-shear ratio. Simultaneous action of shear and axial compression will increase the threshold stiffness by approximately 40% when compared to the plate under pure shear action. Therefore, the safety threshold stiffness formula is suggested, considering the combined action of shear and non-uniform compression.

**Keywords:** steel plate wall; diagonal stiffener; torsional stiffness; elastic buckling; threshold stiffness



## 1. Introduction

Steel plate walls (SPWs), as illustrated in Figure 1, are lateral force resistant elements with great ductility and energy dissipation capability. SPWs are widely used in high-rise and super high-rise buildings [1], such as the Los Angeles Hotel [2] and the 74-storey Tianjin Tower [3]. Thorburn et al. [4] found that thin steel plates formed tension fields after buckling and had a high shear capacity. However, unstiffened SPWs are prone to buckling and exhibit noticeable "pinching" hysteresis curves [5]. Thus, stiffeners are usually applied to the thin SPWs to improve their buckling behaviors [6]. According to certain experiments [7], it was found that the SPWs with flat-bar stiffeners had a considerable influence on the stiffeners after buckling, resulting in twisting and compressive buckling of the stiffeners themselves, which seriously harmed and even failed their stiffening effect. In the Technical Specification for Steel Plate Shear Walls (JGJ/T380-2015) [8], the SPW is designed to bear only horizontal loads, while the SPW will be subject to vertical load such as gravity or live load in actual works. For those reasons, some researchers advocated for the usage of closed form stiffeners to improve the strength and performance of the stiffeners, and allow the SPW to carry a portion of the vertical load [9–11].

In recent years, several researchers have investigated the behavior of SPWs while considering the effect of the torsional stiffness of C-shaped stiffeners. Xu et al. [10,12–14] analyzed the elastic and elastoplastic stability of vertically stiffened SPWs under uniaxial compression, shear, and their combined action. According to these studies, C-shaped

stiffeners with higher flexural and torsional stiffness can significantly increase the elastic buckling stress of stiffened plates. Vu et al. [15] carried out the buckling analysis of transversely stiffened plates under combined bending and shear. The buckling of transversely stiffened corrugated plates was discussed by Feng et al. [16]. Tong and Guo [17] investigated the buckling behavior of vertically stiffened corrugated plates. Furthermore, some experts are concerned about plates that have been diagonally stiffened. Mu and Yang [9] tested the behavior of diagonally stiffened steel plate walls (DS-SPWs) with side openings. Additionally, the shear buckling behavior and threshold stiffness of DS-SPWs with C-shaped stiffeners are studied by the finite element method [18]. Yuan et al. [11], and Martins and Cardoso [19] investigated the elastic shear buckling coefficients for diagonally stiffened plates.

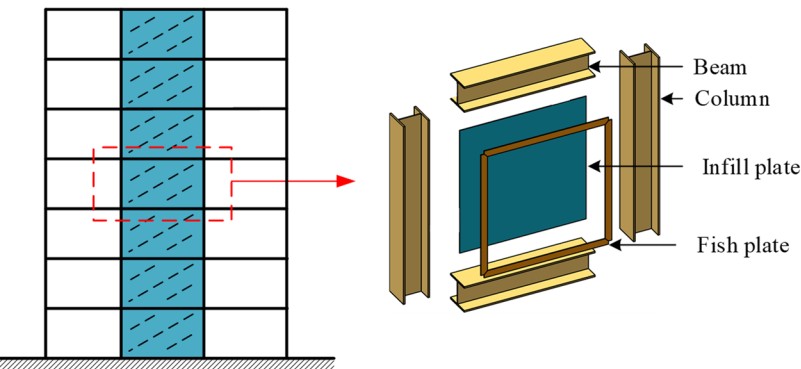

**Figure 1.** Unstiffened steel plate wall.

The current research focuses primarily on the buckling behavior of vertically or diagonally stiffened plates under pure compression or shear. However, there have been few investigations on the diagonally stiffened plate under combined action. In this paper, the influence of the aspect ratio of the plate, the torsional and flexural stiffness of the stiffeners, and a load of combined shear and non-uniform compression on the elastic buckling behavior of the DS-SPWs is investigated. In addition, the formula for the threshold stiffness of the DS-SPWs is established for engineering design, considering the torsional stiffness of stiffeners.

## 2. Elastic Buckling Coefficient and Interaction Formula

### 2.1. Elastic Buckling Stresses for Unstiffened Plate

Timoshenko and Gee [20] initially researched the elastic stability of unstiffened plates and found that the aspect ratio $\gamma$ significantly has a substantial impact on the critical buckling stress $\tau_{cr}$ of the structure, as stated in Equations (1) and (2). The buckling coefficients for a four-edge simply supported plate under shear are calculated by Equations (3) and (4) [21]. The buckling coefficients of a four-edge simply supported plate under bending are approximated by Equation (5) [22].

$$\gamma = \frac{L}{H} \tag{1}$$

$$\tau_{cr} = \frac{k_{cr}\pi^2 D}{H^2 t}, \text{where } D = \frac{Et^3}{12(1-\nu^2)}, \nu = 0.3 \tag{2}$$

$$k_{cr} = 5.34 + 4/\gamma^2 \quad (L \geq H) \tag{3}$$

$$k_{cr} = 4 + 5.34/\gamma^2 \quad (L \leq H) \tag{4}$$

$$\begin{cases} k_{cr} = 15.87 + 1.87/\gamma^2 + 8.6\gamma^2 & (\gamma < 2/3) \\ k_{cr} = 23.9 & (\gamma \geq 2/3) \end{cases} \tag{5}$$

where $L$, $H$, and $t$ is the length, height, and thickness of the plate, respectively; $D$ is the flexural rigidity of the plate; $E$ is elastic modulus of the plate; $k_{cr}$ is the elastic buckling coefficient; and $\nu$ is Poisson's ratio.

### 2.2. Interaction Formula of Unstiffened Plate

Tong [23] analyzed the buckling stability of a four-edge simply supported plate under combined shear and non-uniform compression, giving the interaction formula for plate combined action of axial compression, bending, and shear as Equation (6). The non-uniform compression can be split into a combined action of bending and axial compression. The non-uniform compression distribution factor $\zeta$ and the compression-to-shear ratio $\delta$ are defined in Equations (8) and (9), respectively. Then, the elastic buckling coefficient $k$ of the plate under combined action can be solved by Equation (10), which is obtained from Equations (2) and (6)–(9).

$$\frac{\sigma}{\sigma_{cr}} + \left(\frac{\sigma_b}{\sigma_{bcr}}\right)^2 + \left(\frac{\tau}{\tau_{cr}}\right)^2 = 1 \tag{6}$$

$$\begin{cases} \sigma = (1 - 0.5\zeta)\sigma_{max} \\ \sigma_b = 0.5\zeta\sigma_{max} \end{cases} \tag{7}$$

$$\zeta = \frac{\sigma_{max} - \sigma_{min}}{\sigma_{max}} \tag{8}$$

$$\delta = \frac{\sigma_{max}}{\tau} \tag{9}$$

$$\frac{(1 - 0.5\zeta)\delta k}{k_{\sigma,cr}} + \left(\frac{0.5\zeta\delta k}{k_{\sigma b,cr}}\right)^2 + \left(\frac{k}{k_{\tau,cr}}\right)^2 = 1 \tag{10}$$

### 2.3. Elastic Buckling Coefficient of Diagonally Stiffened Plate in Single Load Action

Mikami et al. [22] used the finite difference method to analyze the buckling behavior of the diagonally stiffened plates. The shear buckling coefficient for the diagonally stiffened plate, which is a four-edge simply supported plate stiffened over the compression and tension diagonal, is given as Equation (11). Moreover, the buckling coefficient of the diagonally stiffened plate in bending and axial compression can be calculated by Equations (12) and (13), respectively.

$$\begin{cases} k_{cr} = 11.9 + 10.1/\gamma + 10.9/\gamma^2 & \text{compression} \\ k_{cr} = 17.2 - 22.5/\gamma + 16.7/\gamma^2 & \text{tension} \end{cases} \tag{11}$$

$$k_{cr} = 22.5 + 4.23\gamma + 2.75\gamma^2 \tag{12}$$

$$k_{cr} = -1.2 + 5.5\gamma + 5.5\gamma^2 \tag{13}$$

## 3. Description of Numerical Model

A diagonally stiffened plate with C-shaped stiffeners, as illustrated in Figure 2, where $b$ is the height of the flange of C-shaped stiffener, $b_s$ is the width of the web of C-shaped stiffener, and $t_s$ is the thickness of C-shaped stiffener.

### 3.1. Definition of Parameters

The stiffener-to-plate flexural stiffness ratio $\eta$ is used to measure the relationship between the flexural stiffness of the stiffener and the plate, as shown in Equation (14). The stiffener's torsional-to-flexural stiffness ratio $K$ is defined as Equation (15). When the C-shaped stiffeners are arranged on both sides of the plate, the moment of inertia $I_s$ and torsional constant $J_s$ of the stiffeners are calculated by Equations (16) and (17), respectively. Thus, $K$ in Equation (15) is a function with factors of $b_s$ and $b$. In this paper, $b_s$ is taken as a constant of 100 mm, and $K$ is changed by varying $b$, as shown in Table 1.

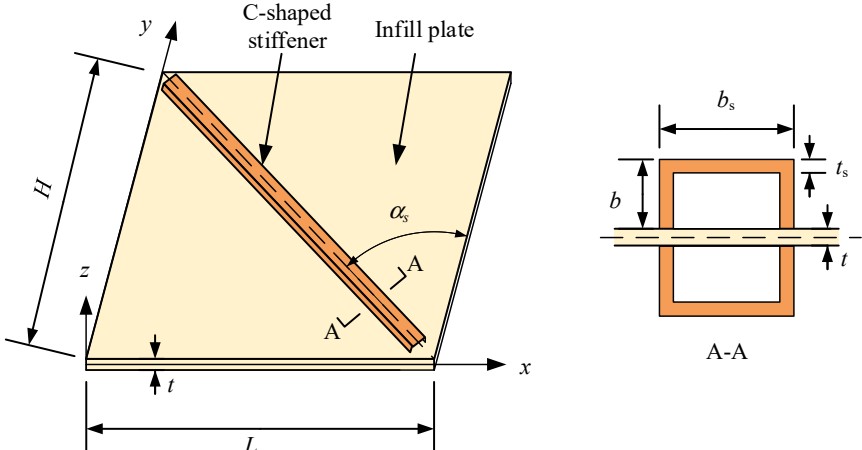

**Figure 2.** Diagonally stiffened steel plate walls.

$$\eta = \frac{E_s I_s}{D L_e}, \text{ where } L_e = H \sin \alpha_s + L \cos \alpha_s \tag{14}$$

$$K = \frac{G_s J_s}{E_s I_s} = \frac{4 b_s^2}{2(1+v)(b_s + 2b)\left(b_s + \frac{2b}{3}\right)} \tag{15}$$

$$I_s = \frac{2 b^3 t_s}{3} + b^2 b_s t_s \tag{16}$$

$$J_s = 2 \frac{b^2 (2b_s)^2 t_s^2}{2 b t_s + b_s t_s} \tag{17}$$

where $\alpha_s$ is the inclination angle of the plate's diagonal; $G_s$ and $E_s$ are the shear modulus and elastic modulus of the stiffeners, respectively.

**Table 1.** The Values of Stiffener's Torsional-to-Flexural Stiffness Ratio $K$.

| Parameter | Value | | | | | |
|---|---|---|---|---|---|---|
| $b/\mathrm{mm}$ | 25 | 50 | 75 | 100 | 125 | 150 |
| $K$ | 0.879 | 0.577 | 0.410 | 0.307 | 0.240 | 0.192 |

This paper investigates the effects of four parameters on the buckling behavior of the diagonally stiffened plates, including the non-uniform compression distribution factor $\zeta$, the compression-to-shear ratio $\delta$, the stiffener's torsional-to-flexural stiffness ratio $K$, and the stiffener-to-plate flexural stiffness ratio $\eta$. SPWs will inevitably be subjected to vertical pressure in actual projects [24], but still dominated by the shear load. Therefore, $\zeta$ ranges from 0 to 1, $\delta$ ranges from 0 to 2, $\eta$ ranges from 1 to 70, and $K$ is taken according to Table 1.

### 3.2. Finite Element Model

Three finite element models are established, as displayed in Figure 3. An unstiffened SPW is used as a validation model in Figure 3a. The stiffened edge of the DS-SPW is assumed to be simply supported Figure 3b. When the stiffeners are rigid, the out-of-plate displacement of the stiffened edge is constrained, and the critical buckling stress value of local buckling can be solved by this model. Figure 3c shows a DS-SPW with C-shaped stiffeners, which is used for the parametric analysis.

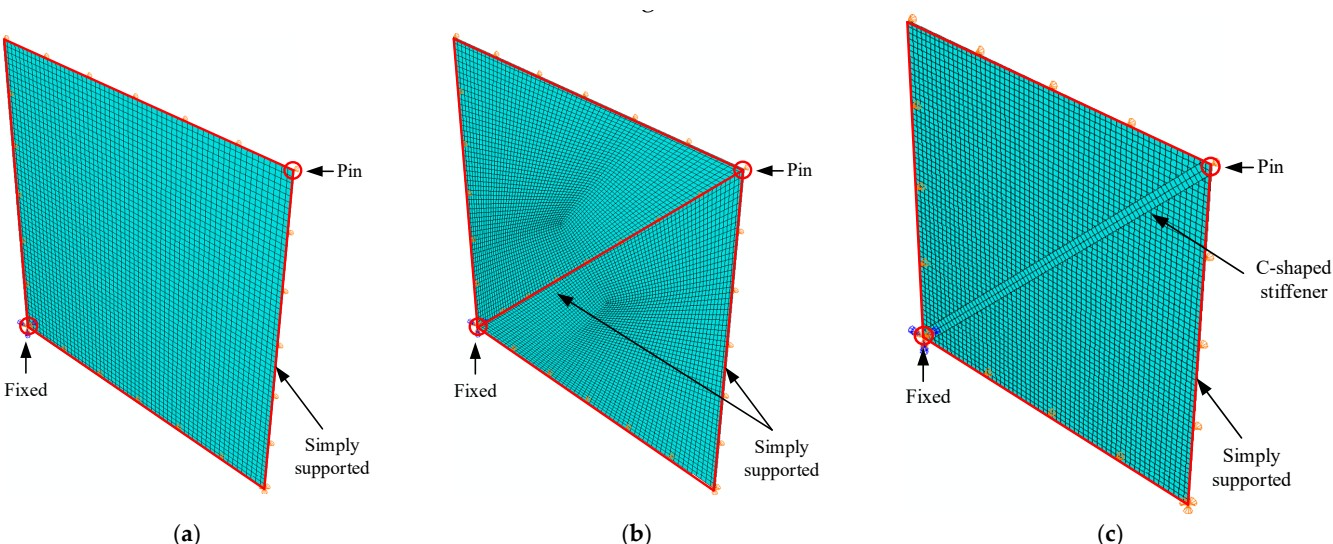

**Figure 3.** Finite models of SPWs: (**a**) SPW, (**b**) SPW with diagonal simply supported, and (**c**) SPW with C-shaped stiffeners.

The plate and the stiffeners are both made of shell element S4R, which is suitable for analyzing the buckling problems of thin panels with out-of-plane deformations. The stiffeners are connected to the steel plate by TIE constraints, which are simulated as a welded connection. To avoid rigid body displacement of the plate, one corner of the plate is fixed and another corner is simply supported. Out-of-plane deformation of four edges is constrained to simulate the case of simply supported. The size of the plate is 3000 mm × 3000 mm × 10 mm ($L \times H \times t$), the size of the C-shaped stiffener is 100 mm × 50 mm × 6 mm ($b_s \times b \times t_s$), and the mesh grid is 50 mm × 50 mm to ensure accuracy and less time-consuming calculation. The modulus of elasticity of the steel is 206,000 MPa, and Poisson's ratio is 0.3. Tangential line loads are applied on the four edges of the plate to simulate the shear stress. Normal line loads are applied on the top and bottom edges of the plate to simulate the compressive stress, and the action form of the non-uniform compression stress can be changed by adjusting the distribution function. The buckling eigenvalue analysis is carried out to solve the buckling stress of the structure.

### 3.3. Verification of Boundary Conditions

The plate in an SPW is connected to the boundary elements by the fishplate, and it is usually assumed that the plate is simply supported on four edges. The elastic shear buckling coefficient of the plates can be calculated by the equations in Sections 2.1 and 2.3.

Table 2 presents the results of the FE analysis for the unstiffened plate when it is subjected to shear, axial compression, bending, or their combined action. The theoretical buckling coefficient *k* can be calculated from Section 2, then *k* is substituted into Equation (2) to obtain the theoretical value of critical buckling stress. Moreover, the critical buckling stresses of DS-SPWs under single load action are similarly verified by comparing the results with Equations (11)–(13). The results indicate that the modelling approach is valid and that it can be utilized to analyze the elastic buckling behavior of the stiffened plate.

**Table 2.** Comparison Buckling Stress of Finite Element and Theoretical Formula.

| Force Conditions | Type | $\zeta$ | $\delta$ | $\sigma_{FEA}$ /MPa | $\sigma_{TH}$ /MPa | $\sigma_{TH}/\sigma_{FEA}$ |
|---|---|---|---|---|---|---|
| Shear | Unstiffened | - | 0 | 19.30 | 19.32 | 1.001 |
| Axial compression | Unstiffened | 0 | 1.0 | 8.12 | 8.27 | 1.018 |
| Non-uniform compression | Unstiffened | 1.0 | 1.0 | 16.05 | 16.12 | 1.004 |
| Bending | Unstiffened | 2.0 | 1.0 | 52.83 | 52.90 | 1.001 |
| Combinded shear and non-uniform compression | Unstiffened | 1.0 | 1.0 | 15.99 | 16.16 | 1.011 |
| Combinded shear and non-uniform compression | Unstiffened | 1.0 | 0.5 | 14.37 | 14.43 | 1.004 |
| Shear (compression) | Diagonally stiffened | - | 0 | 63.6 | 68.0 | 1.069 |
| Shear (tension) | Diagonally stiffened | - | 0 | 23.5 | 23.8 | 1.013 |
| Bending | Diagonally stiffened | 2.0 | 1.0 | 64.5 | 63.1 | 0.978 |
| Axial compressoion | Diagonally stiffened | 0 | 1.0 | 19.6 | 20.9 | 1.066 |

Note: $\sigma_{FEA}$ is buckling stress from FE analysis and $\sigma_{TH}$ is critical buckling stress from theoretical equation.

## 4. Interaction Curves for Diagonally Stiffened Plate

Using the model in Figure 3b as the base model for the analysis in this section, shear stress $\tau$ is applied on four edges of the diagonally stiffened plate, and non-uniform compression stress $\sigma = \zeta\tau$ is applied on the top and bottom edges. The critical shear stress $\tau_{cr}$ of the diagonally stiffened plate under combined action is determined by analysis results, and the critical non-uniform compression stress is calculated as $\sigma_{cr} = \zeta\tau_{cr}$. Then, one point on the interaction curve is obtained as $(\tau_{cr}/\tau_{cr0}, \sigma_{cr}/\sigma_{cr0})$, where $\tau_{cr0}$ is the critical stress of the plate under pure shear and $\sigma_{cr0}$ is the critical stress of the plate under pure non-uniform compression. The interaction curves for the diagonally stiffened plate under combined shear and non-uniform compression can be obtained by varying the compression-to-shear ratio $\delta$ and the non-uniform compression distribution factor $\zeta$.

### 4.1. Combined Shear and Axial Compression $\zeta = 0$

The non-uniform compression is equal to axial compression when the factor $\zeta$ is 0. By varying the compression-to-shear ratio $\delta$, the interaction curves of the diagonally stiffened plate under combined shear and axial compression are shown in Figure 4. The aspect ratio $\gamma$ of the plate affects the shape of the curves. The interaction curves with different aspect ratios are close to parabolas. When $\gamma < 1.2$, the curve rises and moves away from the origin point with increasing $\gamma$. When $1.2 < \gamma < 2.0$, the curve reaches the outermost circle and changes insignificantly. When $\gamma > 2.0$, the curve moves closer to the origin point with increasing $\gamma$.

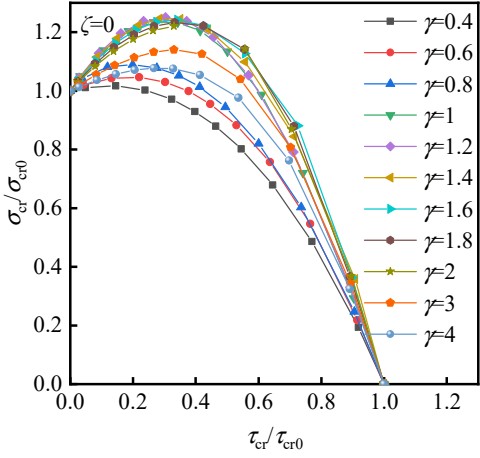

**Figure 4.** Interaction curves of diagonally stiffened plate under combined shear and axial compression $\zeta = 0$.

### 4.2. Combined Shear and Non-Uniform Compression $0 < \zeta < 2$

When the non-uniform compression distribution factor $0 < \zeta < 2$, the interaction curves of the diagonally stiffened plate under combined shear and non-uniform compression are shown in Figure 5. The aspect ratio $\gamma$ and non-uniform compression distribution factor $\zeta$ affect the shape of the curves. The interaction curves with different aspect ratios are also close to parabolas. The variation of the interaction curves with different $\gamma$ is similar to that of factor $\zeta = 0$. As the factor $\zeta$ increases, the peak of the interaction curve gradually decreases, and the axis of symmetry of the parabola is shifted towards the $Y$-axis.

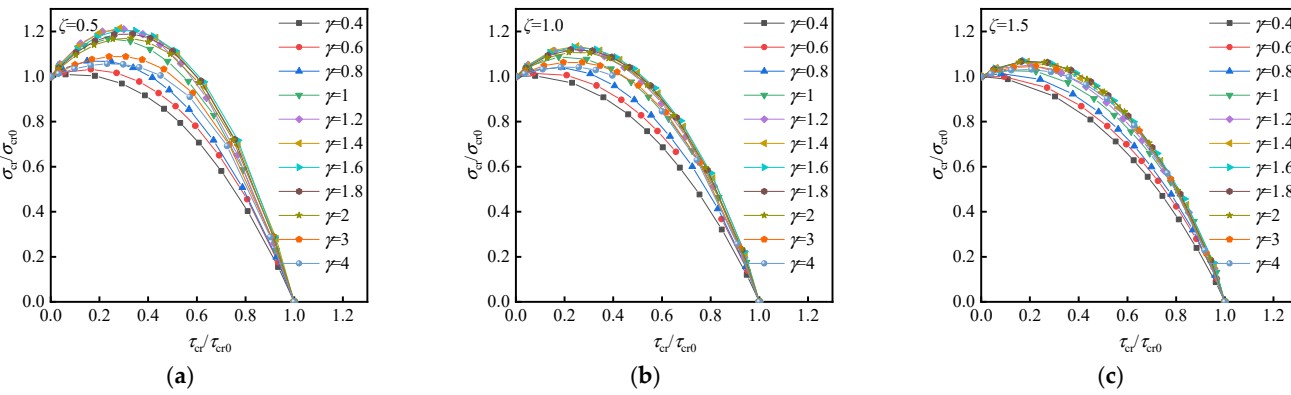

**Figure 5.** Interaction curves of diagonally stiffened plate under combined shear and non-uniform compression: (**a**) $\zeta = 0.5$, (**b**) $\zeta = 1.0$, and (**c**) $\zeta = 1.5$.

### 4.3. Combined Shear and Bending $\zeta = 2.0$

Mikami et al. [22] used Equation (18) to approximate the interaction curves of the diagonally stiffened plate under combined shear and bending when the aspect ratio $\gamma$ of plate was 1.0. Figure 6 illustrated the interaction curves for various aspect ratios. The spacing between each curve is small, and the shape of curves is closer to a parabola curve of which the axis of symmetry is close to the $Y$-axis. There is still a large difference in using Equation (18) to approximate the interaction curve. Therefore, a more accurate approximation curve equation will be given in the following section.

$$\left(\frac{\tau}{\tau_{cr}}\right)^2 + \frac{\sigma_b}{\sigma_{bcr}} = 1, \gamma = 1.0 \tag{18}$$

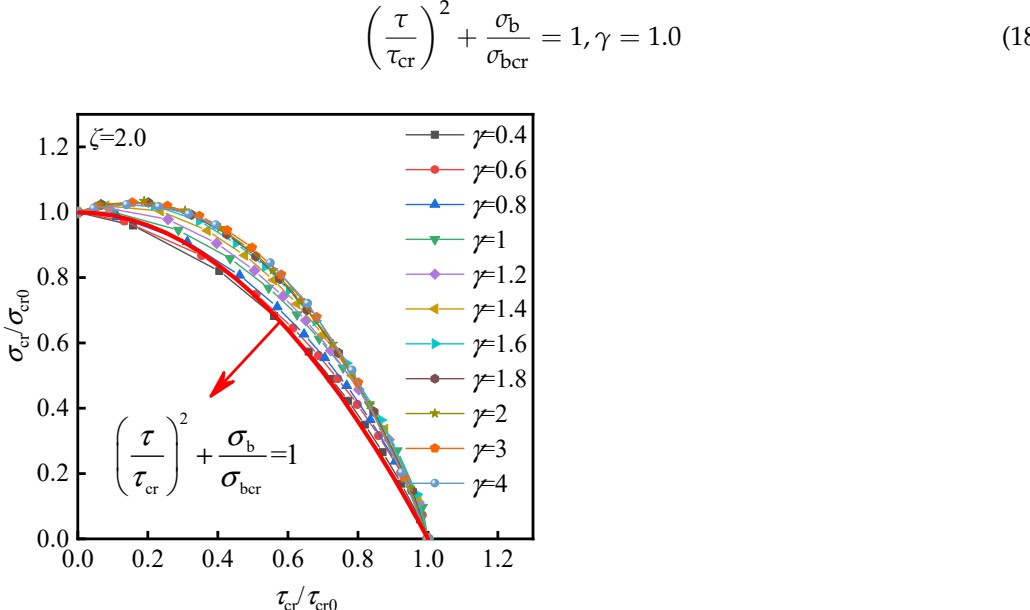

**Figure 6.** Interaction curves of diagonally stiffened plate under combined shear and bending.

### 4.4. Effect of ζ on Interaction Curves

Taking the plate with an aspect ratio $\gamma$ of 1.0 as an example, the interaction curves of the diagonally stiffened plate under combined shear and non-uniform compression are shown in Figure 7. It can be seen that as $\zeta$ increases, the curve gradually converges to a parabola with the axis of symmetry as the Y-axis, and the curve is closer to the origin point. The approximate calculation formula Equation (19) is given according to the variation of the interaction curves. Equation (19) agrees well with the finite element analysis results, indicating that it can reflect the relationship of the diagonally stiffened plate under combined shear and non-uniform compression, i.e., the combined action of shear-bending-axial compression.

$$\left[\frac{5}{2} - \frac{2}{3}\zeta\right]\left(\frac{\tau}{\tau_{cr}}\right)^2 - \left[\frac{3}{2} - \frac{2}{3}\zeta\right]\left(\frac{\tau}{\tau_{cr}}\right) + \frac{\sigma}{\sigma_{cr}} + \left(\frac{\sigma_b}{\sigma_{bcr}}\right)^{\frac{5}{4}} = 1, \gamma = 1.0 \tag{19}$$

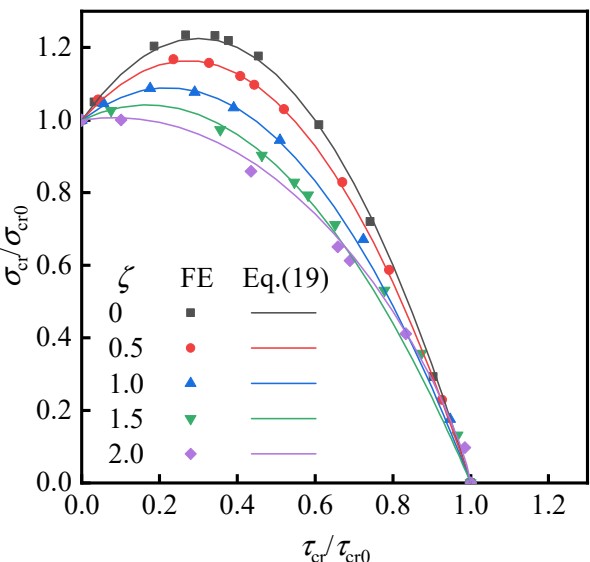

**Figure 7.** Interaction curves of diagonally stiffened plate under combined shear, axial compression, and bending ($\gamma$ = 1.0).

Equation (19) can be written as Equation (20), where $k_{\tau,cr}$, $k_{\sigma,cr}$, and $k_{\sigma b,cr}$ are calculated by Equations (11)–(13). The elastic buckling coefficient $k$ of the diagonally stiffened plate under combined shear and non-uniform compression can be calculated by solving the one-variable function of Equation (20).

$$\left[\frac{5}{2} - \frac{2}{3}\zeta\right]\left(\frac{k}{k_{\tau,cr}}\right)^2 - \left[\frac{3}{2} - \frac{2}{3}\zeta\right]\left(\frac{k}{k_{\tau,cr}}\right) + \frac{(1 - 0.5\zeta)\delta k}{k_{\sigma,cr}} + \left(\frac{0.5\zeta\delta k}{k_{\sigma b,cr}}\right)^{\frac{5}{4}} = 1, \gamma = 1.0 \tag{20}$$

When $\delta = 0$, the diagonally stiffened plate is under the pure shear, at which point the theoretical elastic buckling coefficient $k$ is 32.9, as obtained by Equation (11). The result calculated from Equation (20) is also 32.9. When $0 < \delta < 1.0$, varying the value of $\zeta$, the corresponding elastic buckling coefficient $k$ can be found in Table 3.

**Table 3.** Buckling Coefficients $k$ of the Diagonally Stiffened Plates Under Combined Action ($\gamma$ = 1.0).

| $\delta$ | $\zeta = 0$ | $\zeta = 0.5$ | $\zeta = 1.0$ | $\zeta = 1.5$ | $\zeta = 2.0$ |
|---|---|---|---|---|---|
| 0.25 | 25.4 | 26.3 | 27.3 | 28.5 | 30.0 |
| 0.5 | 19.3 | 20.8 | 22.4 | 24.3 | 26.6 |
| 0.75 | 14.8 | 16.5 | 18.4 | 20.7 | 23.5 |
| 1.0 | 11.6 | 13.3 | 15.3 | 17.7 | 20.7 |

## 5. Threshold Stiffness of DS-SPW

Section 4 gives the interaction curves and elastic buckling coefficients of the diagonally stiffened plate in which the out-of-plane displacement of the stiffened edge is restrained. In actual engineering, the stiffener is not an ideally rigid body. Thus, the flexural stiffness of stiffeners is a significant factor in the design of buildings with DS-SPW.

SPWs are inevitably subjected to shear, axial compression, and bending in the building structures. Their buckling behaviors can be significantly different from those under pure shear action. The performance of the diagonally stiffened plate under pure shear has been studied [18]. Furthermore, the effects of the torsional and flexural stiffnesses of stiffeners on the buckling behavior of the structure under different combined actions are analyzed in this section.

### 5.1. Buckling Mode of DS-SPW

By changing the modulus of elasticity of the stiffeners, the value of the stiffener-to-plate flexural stiffness ratio $\eta$ is changed without affecting any other parameters. Figure 8 illustrates the curves of $\tau$-$\eta$, and the critical local buckling stress is indicated by the blue horizontal line. The shear buckling stress curves of the plate stiffened over the compression and tension diagonal are not the same. Combined with the first-order buckling modes corresponding to different $\eta$ in Figures 9 and 10, the buckling modes of the tensile and compressive types can be divided into global buckling, local buckling, and global–local interaction buckling.

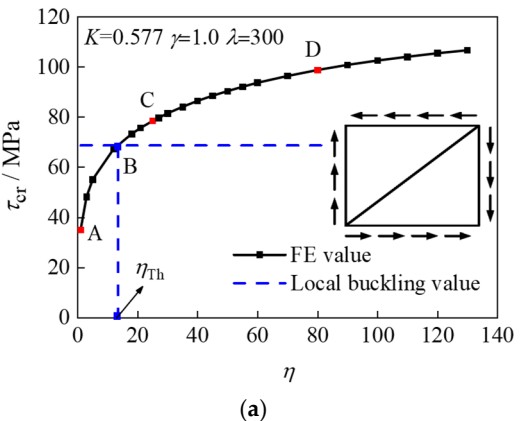
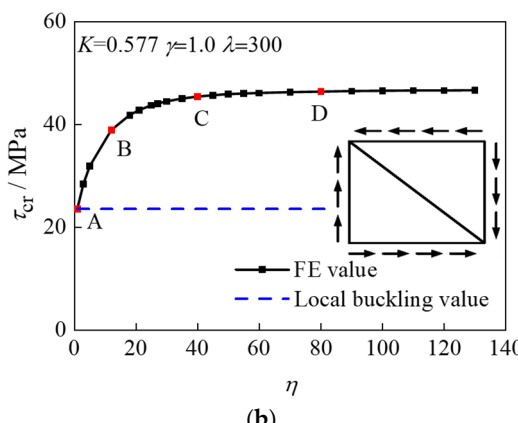

(**a**)                                                                                             (**b**)

**Figure 8.** $\tau$-$\eta$ curve for DS-SPW with C-shaped stiffeners under shear: (**a**) compressive type and (**b**) tensile type.

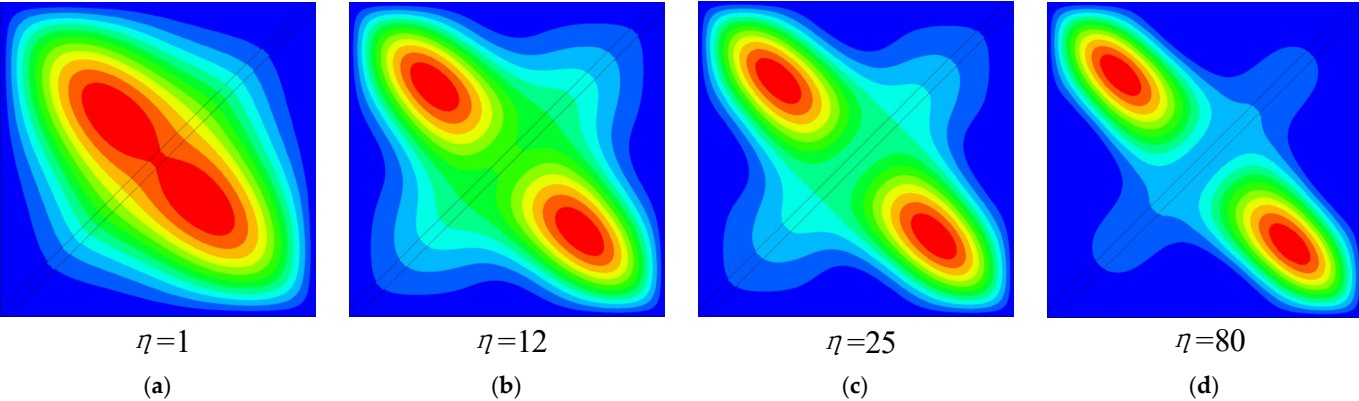

**Figure 9.** Shear buckling mode of SPW with diagonal compression: (**a**) global buckling at Point A, (**b**) local buckling at Point B, (**c**) local buckling at Point C, and (**d**) local buckling at Point D.

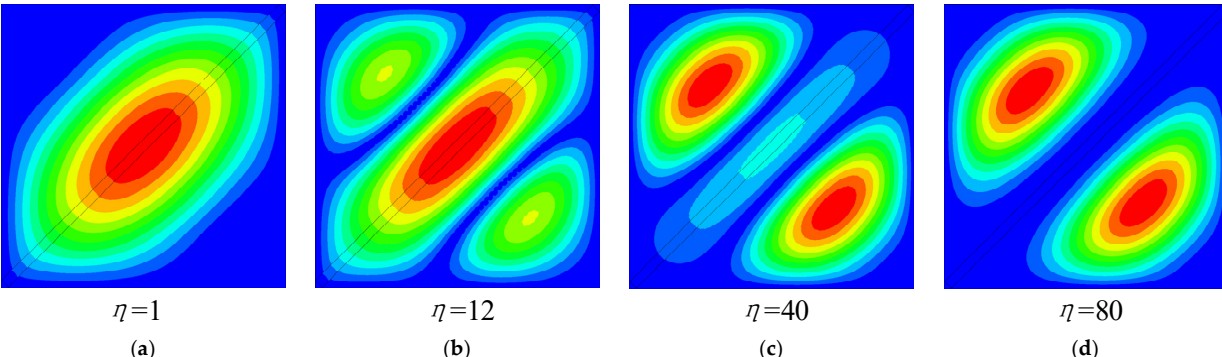

$\eta=1$         $\eta=12$         $\eta=40$         $\eta=80$

(**a**)           (**b**)           (**c**)           (**d**)

**Figure 10.** Shear buckling mode of SPW with diagonal tension: (**a**) global buckling at Point A, (**b**) global–local interaction buckling at Point B, (**c**) local buckling at Point C, and (**d**) local buckling at Point D.

The buckling stress of the plate stiffened over the compression and tension diagonal increases rapidly as $\eta$ increases, such as Figure 9a where the global buckling occurs. When $\eta$ exceeds the critical local buckling load, local buckling of the steel plate occurs, at which time the $\eta$ is called the threshold stiffness $\eta_{TH}$, such as in Figure 9b. It is indicated that the stiffened plate reaches the minimum $\eta$ corresponding to local buckling. As $\eta$ continues to increase, the buckling stress of the structure has increased, and this increased part of value is provided by the stiffeners, such as in Figure 9c,d. Therefore, in a DS-SPW, it is very important to find out the appropriate threshold stiffness $\eta_{TH}$ to ensure that the global buckling of the structure does not occur, and also to make an economical choice of stiffeners.

When the $\eta$ is very small, the global buckling of the plate stiffened over the compression and tension diagonal occurs, such as in Figure 10a. With the increase of the $\eta$, the local buckling occurs between the two nodal lines such as Figure 10b. Then, when $\eta$ exceeds a certain value, the nodal lines of the plate with $\gamma = 1.0$ overlap, decreasing from two to one, and the local buckling occurs in sub-plate such as Figure 10c. After that, as $\eta$ continues to increase, the buckling stress of the structure does not increase.

The $\tau$-$\eta$ curves of the diagonally stiffened plate under combined action are presented in Figure 11. Figure 11a shows the plate under combined shear and axial compression with $\delta$ of 0.25, and the key points on the curve are shown in Figure 12. Figure 11b shows the plate under combined shear and bending with $\delta$ of 1.0, and the key points on the curve are shown in Figure 13. The development of $\tau$-$\eta$ curves is similar to the plate stiffened over the compression and tension diagonal under pure shear, and the buckling mode still changes from global buckling to local buckling with increasing $\eta$. Under combined shear and axial compression action, the local buckling half-wave shifts and is not symmetrical along the diagonal line. Under combined shear and bending action, the maximum deformation point shifts to the upper-left corner. The local buckling has only one half-wave and no antisymmetric half-wave. This is due to the fact that the axial tension force on the right of plate affects the formation of the local buckling half-wave.

## 5.2. Buckling Stress of Stiffened Plate under Combined Action

Changing the modulus of elasticity of the stiffeners and the height of flange of the C-shaped stiffener $b$ varied the value of $K$ without changing other parameters. The C-shaped stiffener has a large torsional stiffness, which obviously affects the elastic buckling stress of the stiffened plate. Figure 14 shows the normalized buckling stress $\tau_{cr}/\tau_{cr0}$ corresponding to different $K$, $\zeta$, and $\delta$. The following regularities can be obtained from Figure 14: (1) When $\zeta$ and $\delta$ remain constants, the buckling stress of the structure keeps increasing as $K$ increases. (2) When $\zeta$ remains constant, the buckling stress of the structure keeps decreasing as $\delta$ increases. (3) When $\delta$ remains constant, the buckling stress of the structure keeps increasing as $\zeta$ increases. In brief, the $K$ and $\zeta$ are positive for improving the elastic buckling stress of stiffened plate, while $\delta$ is negative.

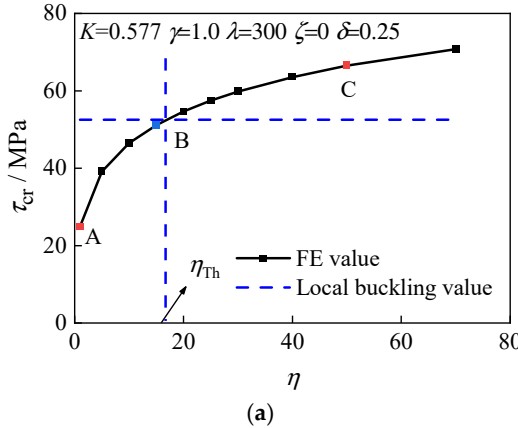 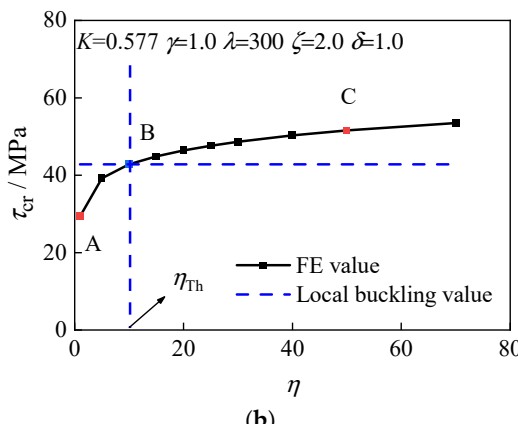

(**a**)  (**b**)

**Figure 11.** $\tau$-$\eta$ curve for DS-SPW with C-shaped stiffeners under combined action: (**a**) $\zeta = 0$ $\delta = 0.25$ and (**b**) $\zeta = 2.0$ $\delta = 1.0$.

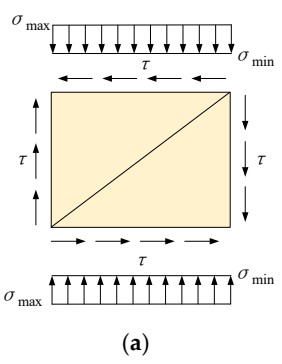 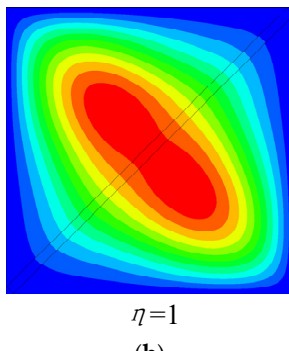 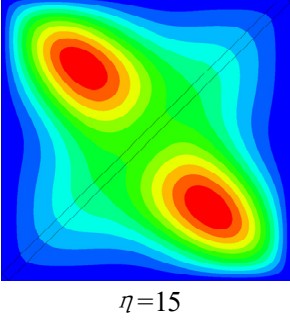 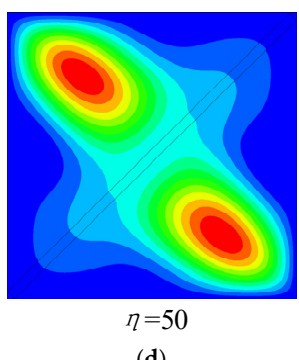

(**a**)  (**b**)  (**c**)  (**d**)

**Figure 12.** Combined shear and axial compression buckling mode of SPW with diagonal compression: (**a**) $\zeta = 0$ $\delta = 0.25$, (**b**) global buckling at Point A, (**c**) local buckling at Point B, and (**d**) local buckling at Point C.

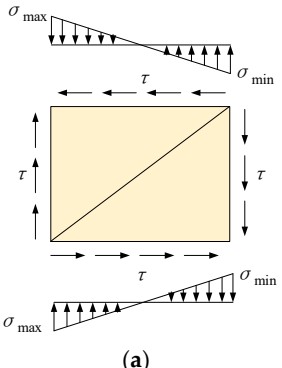 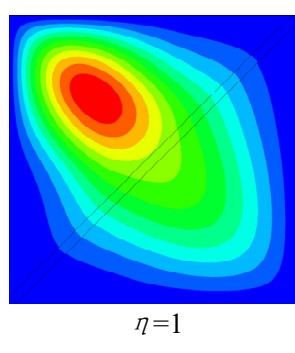 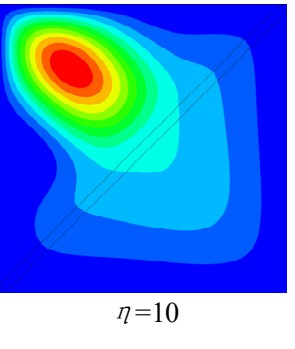 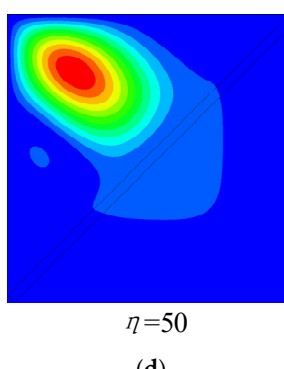

(**a**)  (**b**)  (**c**)  (**d**)

**Figure 13.** Combined shear and bending buckling mode of SPW with diagonal compression: (**a**) $\zeta = 2.0$ $\delta = 1.0$, (**b**) global buckling at Point A, (**c**) local buckling at Point B, and (**d**) local buckling at Point C.

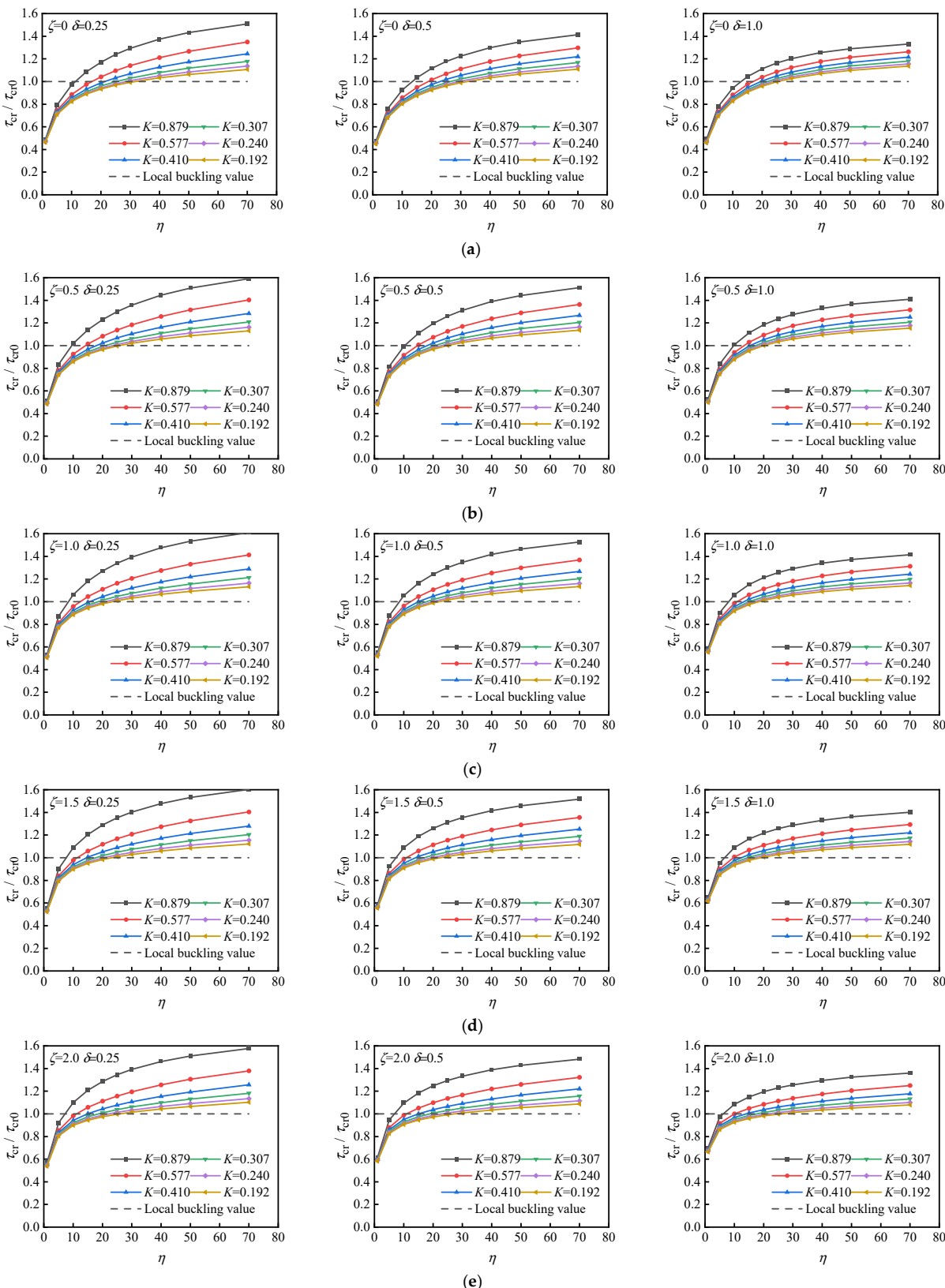

**Figure 14.** $\eta$-$\tau_{cr}$ curves of DS-SPW ($\gamma = 1.0$): (**a**) $\zeta = 0$, (**b**) $\zeta = 0.5$, (**c**) $\zeta = 1.0$, (**d**) $\zeta = 1.5$, and (**e**) $\zeta = 2.0$.

### 5.3. Threshold Stiffness

The intersection of the critical buckling stress and the $\eta$-$\tau_{cr}$ curve is the threshold stiffness $\eta_{TH}$. Then, the threshold stiffness corresponding to different $K$, $\zeta$, and $\delta$ is shown in Figure 15.

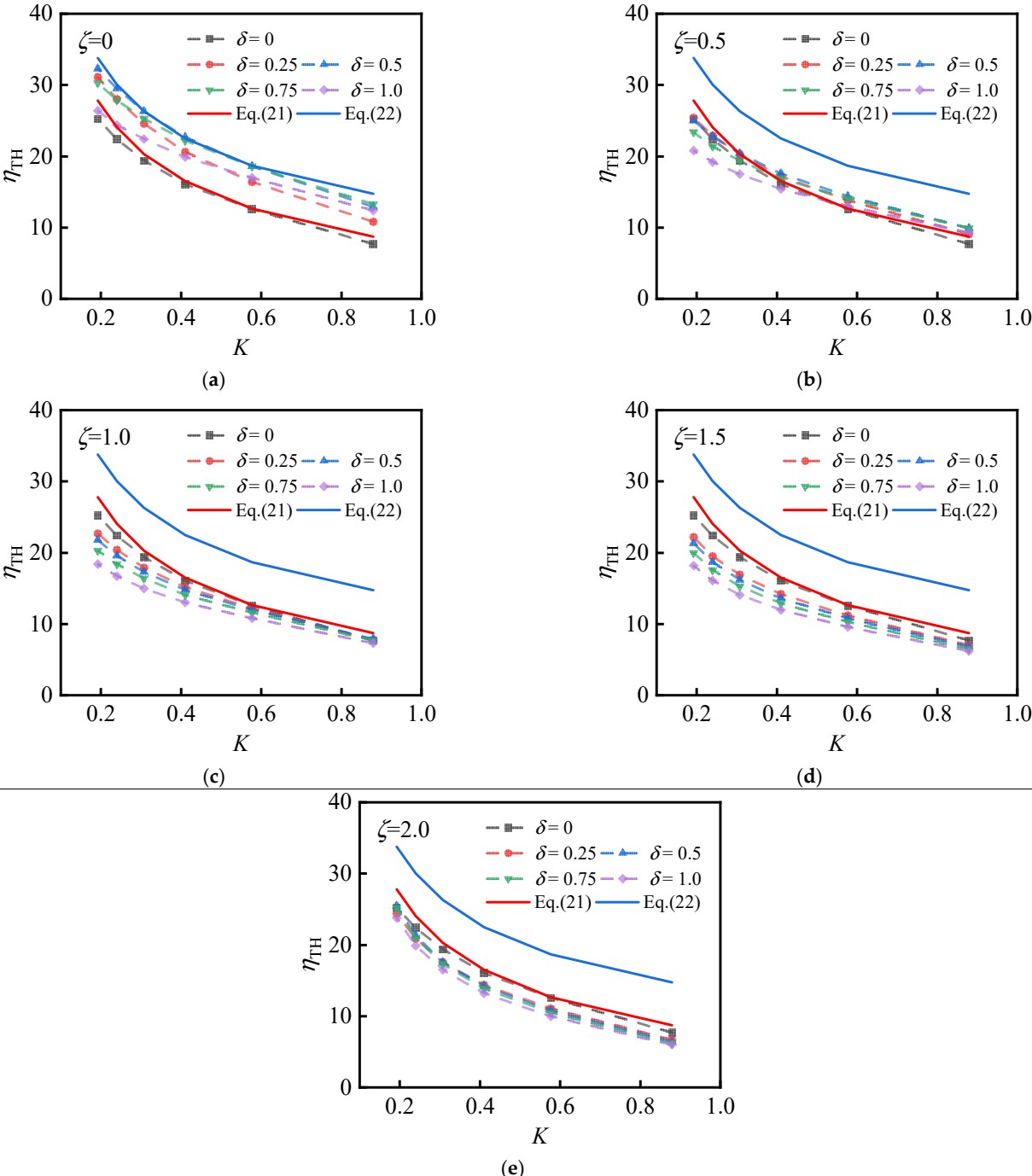

**Figure 15.** $K$-$\eta_{TH}$ curves of DS-SPW ($\gamma = 1.0$): (**a**) $\zeta = 0$, (**b**) $\zeta = 0.5$, (**c**) $\zeta = 1.0$, (**d**) $\zeta = 1.5$, and (**e**) $\zeta = 2.0$.

With the increase of $K$, the threshold stiffness tends to decrease, indicating that increasing the $K$ of the stiffeners can reduce the requirement of flexural stiffness of the stiffeners. The height of the stiffeners is higher with a lower value of $K$, so the stiffeners themselves are more likely to twist. Therefore, a higher flexural stiffness of the stiffener is required. According to the analysis under pure shear, the $K$ of the stiffener is recommended to be not less than 0.3 [18].

Using $\delta = 0$ as a reference line (i.e., the threshold stiffness of the stiffened plate under pure shear), as $\delta$ increases, the threshold stiffness of the structure does not change in the same trend. When $\zeta = 0$, the threshold stiffness is maximum at $\delta$ of 0.5. That is, the threshold stiffness of the structure is maximum when the plate is under combined shear and axial compression. At this time, the threshold stiffness is increased by about 40% compared to the pure shear state. Therefore, if an SPW is subjected to a combination of shear and axial compression in practice, the threshold stiffness obtained from plate under pure shear is on the low side and the structure will occur global buckling instead of local buckling. The threshold stiffness curve of the combined action tends to decrease as $\zeta$ increases. When $\zeta = 0.5$, it basically overlaps with the threshold stiffness curve of pure shear. When $1.0 < \zeta < 2.0$, the threshold stiffness curve of joint action is below the threshold stiffness curve of pure shear. In this case, taking the plate under pure shear as the threshold stiffness is on the safe side.

The most unfavorable action is the combination of axial compression and shear. The threshold stiffness requirement can be satisfied by considering this case only during the design. For diagonally stiffened plates in pure shear or $1.0 < \zeta < 2.0$, the threshold stiffness can be calculated according to Equation (21). For $0 < \zeta < 1.0$, the threshold stiffness can be calculated according to Equation (22), and the results obtained are safe.

$$\eta_{TH} = -8 + 15.7/K^{0.5}, \zeta = 0 \tag{21}$$

$$\eta_{TH} = -2 + 15.7/K^{0.5}, \zeta \leq 1.0 \tag{22}$$

## 6. Conclusions

In this study, a diagonally stiffened steel plate wall (DS-SPW) with C-shaped stiffeners is presented and elastic eigenvalue buckling calculations are performed using ABAQUS software to analyze the buckling stresses and buckling modes of DS-SPW under combined shear and non-uniform compression. The accuracy of the performance of the numerical model is first verified with the four-sided simply supported rectangular plate and the diagonally stiffened plate. The parameters, such as the compression-to-shear ratio, the non-uniform compression distribution factor and the stiffener's torsional-to-flexural stiffness ratio, and the elastic buckling performance of the stiffened plates, are changed to carry out a large number of numerical models for DS-SPWs, and the elastic buckling performance of diagonally stiffened plates under combined shear and non-uniform compression is investigated. Considering the torsional stiffness of C-shaped stiffeners, the formula of threshold stiffness, which is the minimum stiffness of stiffeners required for a stiffened plate from overall buckling to local buckling, is proposed. The main results of the study are as follows.

(1) The elastic buckling behavior of the diagonally stiffened plate under combined shear and non-uniform compression is analyzed. The interaction curves of a plate under combined action are parabolic, and the aspect ratio $\gamma$ and the non-uniform compression distribution factor $\zeta$ obviously affect the shape of the parabolas. The larger the factor $\zeta$ is, the more the parabola moves toward the origin point and the axis of symmetry of the parabola is closer to the $Y$-axis. As the aspect ratio increases $0.4 < \gamma < 1.2$, the interaction curve moves away from the origin point; when the aspect ratio continues to increase $1.2 < \gamma < 2.0$, the interaction curve is closer to the origin point, and the change is not obvious; as the aspect ratio continues to increase $2.0 < \gamma < 4.0$, the curve moves closer to the origin point. When $\gamma = 1.0$, the interaction equations of diagonally stiffened plate combined shear and non-uniform compression, considering the non-uniform compression distribution factor $\zeta$ and the compression-to-shear ratio $\delta$, are given; moreover, the equation of critical elastic buckling coefficient of that are also given;

(2) The effect of stiffener's torsional-to-flexural stiffness ratio $K$ on the elastic buckling stresses of stiffened plates is investigated. With the increase of the stiffener-to-plate

flexural stiffness ratio $\eta$, the buckling stress of the stiffened plate with diagonal tension increases rapidly and then tends towards a certain value. The buckling stress of the stiffened plate with diagonal compression increases with $\eta$, which is due to the diagonal stiffeners assuming the role of compression bars. As $K$ and $\zeta$ increase, the buckling stress of the structure increases, whereas, an increase in $\delta$ decreases the buckling stress of the structure;

(3) The effects of $\zeta$, $K$, and $\delta$ on the threshold stiffness $\eta_{TH}$ are analyzed. The threshold stiffness of the structure decreases when $K$ and $\zeta$ increase. When $1.0 < \zeta < 2.0$, increasing $\delta$ can reduce the threshold stiffness of the structure, at which time the threshold stiffness under combined action is lower than that under pure shear action. When $\zeta = 0$, the threshold stiffness of the structure under the combined axial compression and shear is 40% higher than that under pure shear. At this time, if the threshold stiffness in pure shear will not ensure that the structure meets the requirements of the critical local buckling stress, the global buckling will occur and it is unsafe. Therefore, the calculation of threshold stiffness is given in favor of safety.

**Author Contributions:** Conceptualization, Y.Y. and Z.M.; methodology, Y.Y.; software, Y.Y.; validation, Y.Y.; formal analysis, Y.Y.; investigation, Y.Y.; resources, Y.Y., Z.M. and B.Z.; data curation, Y.Y.; writing—original draft preparation, Y.Y.; writing—review and editing, Z.M. and B.Z.; visualization, Y.Y.; supervision, Z.M. and B.Z.; project administration, Z.M.; funding acquisition, Z.M. All authors have read and agreed to the published version of the manuscript.

**Funding:** This research was funded by [National Natural Science Foundation of China] grant number [51578064]; [Natural Science Foundation of Beijing Municipality] grant number [8172031].

**Institutional Review Board Statement:** Not applicable.

**Informed Consent Statement:** Not applicable.

**Data Availability Statement:** All the data supporting the results were provided within the article.

**Conflicts of Interest:** The authors declare no conflict of interest.

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
