# Peer review of "Numerical Study on Elastic Buckling Behavior of Diagonally Stiffened Steel Plate Walls under Combined Shear and Non-Uniform Compression"

_metals, doi:10.3390/met12040600_

Round 1

Reviewer 1 Report

Remarks:
There are no other references to significant publications on the shear buckling factors of a reinforced plate. 

  1. How was the C-channel connected to the steel plate, were contact elements used? How can this be realized in real constructions?
  2. Why was the S4R element used, can other better-fit finite elements be used?
  3. Line 138 - Why a 50x50mm finite element mesh was chosen?
  4. What was the method of testing the convergence of the solution? (deformation, stress, strain or other) 
  5. It would be good to verify theoretical and numerical considerations with experimental research in the future
  6. Figure 9 and others (Line 253) - Why unsymmetrical local buckling forms appear?

Reviewer 2 Report

The paper was written in a very thoughtful manner. The research topic has been presented thoroughly and the content of the paper is not objectionable. After reviewing the content of the work and the enormous contribution of the authors to its creation, I conclude that the work can easily be published in its current form, due to the quality and manner of the content and research results presented in it.

Reviewer 3 Report

Dear Author

Your paper consider only numerical researches so I suggest changing the title. As for the main text of paper it is written well, but because in consist only numerical test it is hard to say how close to the reality are obtained equations. As for figure 14 this is completely impossible to read and analyze, too small and too many in one place. In the paper there authors do not placed any summary. In my opinion this should be placed before conclusions.            

Round 2

Reviewer 3 Report

In this form the paper can be pulished